# Contextuality in Collective Intelligence: Not There Yet

**DOI:** 10.3390/e25081193

**Published:** 2023-08-11

**Authors:** William Sulis, Ali Khan

**Affiliations:** 1Collective Intelligence Laboratory, McMaster University, 92 Bowman St., Hamilton, ON L8S 2T6, Canada; 2Department of Psychology, Neuroscience an Behaviour, McMaster University, 1280 Main St W., Hamilton, ON L8S 4K1, Canada; ali.khan0@gmail.com

**Keywords:** contextuality, collective intelligence, intransitive decision making, social insects

## Abstract

Type I contextuality or inconsistent connectedness is a fundamental feature of both the classical as well as the quantum realms. Type II contextuality (true contextuality or CHSH-type contextuality) is frequently asserted to be specific to the quantum realm. Nevertheless, evidence for Type II contextuality in classical settings is slowly emerging (at least in the psychological realm). Sign intransitivity can be observed in preference relations in the setting of decision making and so intransitivity in decision making may also yield examples of Type II contextuality. Previously, it was suggested that a fruitful setting in which to search for such contextuality is that of decision making by collective intelligence systems. An experiment was conducted by using a detailed simulation of nest emigration by workers of the ant *Temnothorax albipennis*. In spite of the intransitivity, these simulated colonies came close to but failed to violate Dzhafarov’s inequality for a 4-cyclic system. Further research using more sophisticated simulations and experimental paradigms is required.

## 1. Introduction

In spite of a century of computational success, quantum mechanics still lacks a consistent interpretation, although there have been many proposals over the years. This has lent quantum mechanics an air of inscrutability and the pervasive notion that the quantum and classical realms constitute two solitudes. Nevertheless, there are aspects of quantum mechanics, particularly that of contextuality, which have attracted the interest of serious researchers, particularly within the life and social sciences, and which may in fact cross over into the classical realm. In [1], I suggested that part of the reason why it has been so difficult to find a consensual interpretation is a bias, implicit or explicit, towards what I have termed the Objectivist Worldview. The central entity within the Objectivist Worldview is the *object*. The prime examples of objects are found in mathematics. The central entities of study within quantum mechanics (and physics generally) are formed of inanimate matter, which come closest to embodying the characteristics of an ideal object. The principal characteristics of an (ideal) object include the following:It exists independent from any other entity—it can be isolated and treated as a whole unto itself.Entanglement does not occur.It is eternal—it does not *become*, it merely *is*.It is passive—it reacts, it does not act.Its properties are intrinsic and noncontextual—they are fixed, complete and independent of the actions of any other entity.Its motion is determined by fixed laws—which may be deterministic or stochastic (usually explained away as due to ignorance on the part of the observer).Its motion is often attributed to variational principles—optimality, minimal and maximal—always extremized in some direction.Its interactions with other objects are always local.History is irrelevant—the future motion of an object depends only on its present state (and sometimes not even that in the case of stochastic objects).

Unlike classical systems, quantum mechanical systems frequently fail to exhibit one or more of these characteristics. I believe that the mismatch between an implicit Objectivist Worldview and the characteristics of quantum mechanical systems contributes to the inability to find a consistent interpretation of quantum mechanics.

Living and social systems, on the other hand, possess a very different set of characteristics which render them nearly in opposition to the ideal of an object. Some of these characteristics are as follows:They are embedded in an environment, physical or social—they cannot be isolated and treated as a whole unto themselves.Entanglement is commonplace.They are transient—they come into existence, persist for some duration, then fade away.They are active—they act upon their environment and do not merely react to it.Their properties are contextual—their determination requires interaction with other entities, and such interactions exhibit exclusions and order effects.Their motions are determined more by rules and influences rather than by fixed laws—information plays a central role.Variational principles may play a role in some circumstances.Interactions with other objects are always causal but may be nonlocal through the intermediary of information-laden signs.History is fundamental—the future motion of an object depends on its history, not merely its present state.

These characteristics are more typical of the concept of process, which forms the basis for the Processist Worldview.

I have suggested [1,2] that quantum systems have much more in common with biological and social systems than with inanimate objects and that a shift to a Processist Worldview [1] might lead to more effective interpretations, restoring a sense of reality to physics (though not an Objectivist reality) and resolving the quantum–classical dichotomy. The Processist Worldview posits that entities are generated by processes (exemplified by biological organisms) and that processes are generative, contextual, transient (becoming and fading away are fundamental), non-Kolmogorov, active, interactive and driven by information and functionality. Inspired by the writings of Robert Rosen [3], I have approached quantum mechanics from the perspective of complex systems theory, asking what concepts and methods from the study of complex systems (e.g., [4]) might be fruitfully applied to shed light onto the fundamental problems of quantum mechanics. Further inspired by Whitehead [5] and Trofimova [6,7,8], I was led to the idea of Process Algebra, which provides a formal framework to describe and model these features of process and which has been applied with some success to the development of realist models of nonrelativistic quantum mechanics [9,10,11,12].

If, as I have suggested, quantum systems and biological and social systems share many characteristics, then it is reasonable to ask which features of quantum theory might be fruitfully applied to the study of biological and social systems. Many authors have explored possible connections between these disciplines [13,14,15,16]. One subject which shows considerable promise is contextuality. It has long been known in the life and social sciences that there is no such thing as a disturbance-free measurement. Every measurement introduces a context within which an observed entity must act, and each such context may elicit its own specific actions and probabilities of appearance. Moreover, the order of measurements can affect the results that are obtained. Indeed, some measurements cannot be carried out simultaneously or can only be carried out in certain orders and not in others (try dissecting a cat first and exercising it later!). Contextual effects have been observed in the life sciences [17], decision theory [18] and in the construction of science [19].

This form of contextuality has been termed inconsistent connectedness in the Contextuality by Default approach of Dzhafarov [20,21]; I have called it Type I contextuality [10], and it lies at the heart of Bohr’s Copenhagen interpretation of quantum mechanics. This form of contextuality appears to be ubiquitous among complex systems, though its implications for a description in terms of probability and statistics remains largely unrecognized outside of quantum mechanics. The need to take the context of the measurement into account requires a modification of the usual Kolmogorov probability structure, whether invoking Khrennikov’s Contextual Probability theory [22] or Dzhafarov’s Contextuality by Default model [20]. Contextual Probability theory does for Probability Theory what the discovery of non-Euclidean Geometry did for Geometry in the 19th century. Contextuality by Default takes a different approach, keeping the usual Kolmogorov structure but expanding it by requiring that each random variable be associated with a particular context within which it is defined. The same measurement procedure applied in different contexts may give rise to different random variables for the supposedly same construct. For example, a psychological test applied in different contexts could yield different probability distributions for its test items (i.e., different random variables). Failing to take this into account and simply assuming that they are all the same and combining them into a single joint distribution could result in erroneous results.

A second form of contextuality has received much attention in the quantum mechanical literature. This has been termed true contextuality by Dzhafarov [20] and Type II contextuality by me [10]. This form of contextuality concerns the occurrence of correlations among a set of random variables (measurements) which are not possible within the usual framework of Kolmogorov Probability Theory. These correlations are gathered together in a formula whose value is bounded whenever standard Kolmogorov probabilities are involved and which can violate this bound when quantum-mechanical-type probabilities (the Born rule) are involved. These probabilities are, by definition, non-Kolmogorov. The values for quantum mechanical systems are themselves bounded by the so-called Tsirel’son bound. There are many such formulae and corresponding bounds (expressed in the form of inequalities): Bohm [23], Bell [24], Leggatt–Garg [25], CHSH [26], Gisin [27] and Mann [28], but the most widely studied is the CHSH inequality. Dzhafarov developed an analogue of the CHSH inequality, which finds application in the study of psychological experiments [20].

In quantum mechanics, the explanation for the appearance of these inequality violations is usually attributed to the presence of nonlocal influences, what Shimony termed “Passion at a Distance” [29]. These influences are not only nonlocal but must also be superluminal, as a simple argument shows [9]. Gisin has argued that they must in fact be instantaneous no matter the distance [27]. Much effort has been expended to explain the nature of these influences in the face of the absence of superluminal signaling as required by Special Relativity due to the constancy of the speed of light in inertial reference frames. I particularly like Griffiths’ comment [30]: “To be sure, those who claim that instantaneous nonlocal influences are present in the quantum world will generally admit that they cannot be used to transmit information; this is known as the ‘no-signaling’ principle, widely assumed in quantum information theory. This means that such influences (including wavefunction collapse) cannot be directly detected in any experiment. The simplest explanation for their lack of influence is that such influences do not exist”. There are indeed nonlocal influences in the biological and psychological realms. In neurobiology and collective intelligence systems for example, nonlocal influences are mediated through the volume transmission of neurotransmitters and hormones and by the diffusion of pheromones [2]. These influences are neither superluminal nor instantaneous. Dzhafarov and colleagues have demonstrated the existence of true contextuality in two experiments [31,32,33], as have other authors [34,35,36]. A simple thought experiment involving ice cream preferences shows a violation of even the Tsirel’son bound under ideal conditions [37] (I will describe this in more detail later). In none of these cases are superluminal influences involved.

As several authors have argued, the appearance of these exceptional correlations is not due to nonlocality but rather to the presence of contextuality [30,38,39,40]. The experiments mentioned above are all examples of contextuality, not nonlocality. The attachment to a notion of nonlocality as a cause of these correlations is again a throwback to an implicit Objectivist Worldview. In such a worldview, spatially separated entities must also be statistically independent entities. From a Processist perspective, in which entities are generated by processes, there is no a priori reason why spatially separated entities cannot be statistically dependent since they may be the outcome of the action of a single process. There is nothing a priori to suggest that a single process must only generate single entities. It is perfectly plausible that they could generate multiple entities, and as has been argued elsewhere [1], these processes are generators of space and time, and therefore they themselves exist outside of space and time, an idea suggested by Gisin [27]. They could conceivably generate entities wherever and whenever, so long as relativistic considerations are respected, including the absence of superluminal information propagation.

Mathematics, the epitome of the Objectivist Worldview, has been profoundly influential in shaping our collective view of the world. Mathematics is held out as the embodiment of all that is ideal. For example, Wigner [41] prefaces his famous paper on the unreasonable effectiveness of mathematics in physics with a quote from Bertrand Russell: “Mathematics, rightly viewed, possesses not only truth, but supreme beauty cold and austere, like that of sculpture, without appeal to any part of our weaker nature, without the gorgeous trappings of painting or music, yet sublimely pure, and capable of a stern perfection such as only the greatest art can show. The true spirit of delight, the exaltation, the sense of being more than Man, which is the touchstone of the highest excellence, is to be found in mathematics as surely as in poetry”.

Attributes that are held to be of value in the development of mathematics are thought to be of equal value in the natural world—that the natural world *aspires* to be like the mathematical world, and when it deviates from it, that represents not novelty or creativity but rather failure. A failure to be rational, extremal or symmetrical is to be irrational, suboptimal or distorted. Since the time of Plato, nature has been viewed as a poor simulacrum of the true reality. For all of its beauty, I believe that mathematics, at least in its current form, is unduly restricted, and that nature excels in its ability to cobble together effective solutions in the face of imperfect knowledge, transient, ever-changing conditions/contexts, limitations on resources and never-ending perturbations. These departures from the so-called ideal may be responsible for the contextuality, both Type I and II, that has been observed. It may be precisely these “non-ideal” features which allow classical systems to surpass these bounds and generate stronger correlations than would be obtainable by those “ideal” entities and interactions.

A hint of this may be found in the form of the CHSH inequality. The CHSH inequality is a special case of the following inequality:|f(a,b)+f(b,c)+f(c,d)−f(d,a)|≤2
where a,b,c and *d* are some kind of value, function or even index, and *f* is a real valued function which takes its values in the range [−1,1]. In the case of CHSH, the domain variables represent choices of measurements while *f* is a correlation function. The CHSH scenario is an example of what Dzhafarov terms a 4-cyclic system. We may represent this in the form of a graph:$$\begin{array}{ccc} a & - & d\\ | & & |\\ b & - & c \end{array}$$

Suppose that we assign real values to each vertex, and to each edge we assign the product of the values at the vertices. This gives us
$$\begin{array}{ccc} a & \overset{ad}{-} & d\\ \left(ab)\right| & & \left|(cd\right)\\ b & \overset{-}{bc} & c \end{array}$$

Notice that if ab,bc and cd are all positive in sign, then so must be ad, and likewise in the negative case. There is thus a kind of transitivity for signs in the 4-cyclic case. In the more general case, we say that the 4-cyclic system possesses *f*-sign transitivity whenever sign [f(a,b)]= sign[f(b,c)]= sign[f(c,d)]=s, and then sign[f(a,d)]=s. The case of sign transitivity can be depicted as
$$\begin{array}{ccc} a & \overset{f(a,d)}{\to } & d\\ f(a,b)↓ & & ↑f(c,d)\\ b & \vec{f(b,c)} & c \end{array}$$
which represents a linear or transitive ordering.

Now note that the maximum value of
|f(a,b)+f(b,c)+f(c,d)−f(d,a)|
is four, and this value can only be obtained in one of two ways:f(a,b)=f(b,c)=f(c,d)=1;f(a,d)=−1
or
f(a,b)=f(b,c)=f(c,d)=−1;f(a,d)=1

This corresponds to the diagram
$$\begin{array}{ccc} a & \overset{f(a,d)}{\leftarrow } & d\\ f(a,b)↓ & & ↑f(c,d)\\ b & \vec{f(b,c)} & c \end{array}$$
which represents a cyclic or intransitive ordering. The notion of sign transitivity can be extended to *n*-cyclic systems for arbitrary *n*.

Although not proven, it may be the case that sign intransitivity cannot hold for 4-cyclic systems and indeed for any even-order cyclic system so that such inequalities cannot be maximally violated except perhaps in the presence of inconsistent connectedness. This requires further study. Nevertheless, in the case of odd-order cyclic systems, it may be useful to look for violations of the inequality under conditions in which sign transitivity is broken. Of course, there is no guarantee that merely breaking sign transitivity will result in a violation of the inequality, but the presence of sign transitivity will definitely ensure that the inequality is not maximally violated. One setting in which violations of transitivity occur is in decision making.

### 1.1. Intransitivity in Human Decision Making

Rationality, as a mode of decision making, combines two characteristics of the Objectivist Worldview—logic and optimality. Individuals, when making choices, may show preference for one choice over others or indifference when alternatives may be chosen with equal probability. Rationality in the context of preference is thought to require the presence of transitivity [42], whereby if an agent prefers choice A over choice B, and choice B over choice C, then they will prefer choice A over choice C, regardless of context, order of presentation or the presence of competing attributes. Preferences that are not transitive are said to be intransitive and are viewed as examples of irrational decision making and hence are to be avoided or derided. The presumption of transitivity is also necessary in order for an ordinal scale of preferences to exist [43]—this is useful mathematically, but there is no reason a priori that nature should organize itself in such a manner as to guarantee us nice mathematical models. In the decision literature, two types of transitivity are studied: strong transitivity (ST), which is deterministic, and weak transitivity (WT), which is stochastic, requiring that the preferred choice be chosen more than 50% of the time.

Butler and Progrebna [44] argue that “Transitivity must hold either if a value attaches to each option without reference to other alternatives (choice-set independence), or if an equivalent value results after comparing and contrasting the attributes of the available choice options”. Anand [45] argued that intransitivity need not necessarily be irrational. Bar-Hillel and Margalit [46] argued for three contexts in which intransitivity might reasonably occur:where intransitivity results from the application of an ethical or moral choice rule;where intransitivity results from the application of an ethical or pragmatic choice rule;where the choice is intrinsically comparative, depending upon multiple competing alternatives.

In these contexts, intransitivity presents as a plausible consequence. Tversky [43] demonstrated the occurrence of consistent and predictable intransitivity in certain situations of decision making. Since then, there have been many studies demonstrating the occurrence of intransitivity in decision making [47,48]. Nevertheless, several authors continue to deny the existence of intransitive preferences [49,50,51,52]. Regenwetter, Dana and Davis-Stober claim that transitivity is in fact a universal phenomenon and any appearance of intransitivity is a sign of methodological error [49,50].

Consider the following example based on personal experience. I judge ice cream based on two attributes: taste and the propensity to cause gastroesophageal reflux. One attribute provides pleasure, and the other pain. Let us consider four types of ice cream together with their attribute ratings (taste and reflux potential) based on a sliding scale from 0–100: double-fudge chocolate (100,100), double chocolate (75,75), chocolate (50,50) and pistachio (25,25). I will choose pleasure over pain, but only to a certain point. If the difference in reflux potential between two alternatives is 50 or less, I choose by taste. If the difference in reflux potential exceeds 50, then I prefer to minimize pain, choosing whichever has the lower reflux potential. This gives a set of preferences in the form of double-fudge chocolate > double chocolate; double chocolate > chocolate; chocolate > pistachio; pistachio > double-fudge chocolate. This preference relation is clearly intransitive, yet it is principled, and I would argue rational given my tolerances. If my choices are consistently applied, then it is possible to construct a set of random variables on this set of preferences such that the corresponding CHSH inequality is maximally violated, exceeding both the quantum mechanical and Tsirel’son bounds by taking a value of four.

Note that a different set of preferences will occur if I am given three choices simultaneously. For example, out of double-fudge chocolate, double chocolate and pistachio, I would choose pistachio since the first two are guaranteed to cause a lot of reflux. Yet, given double-fudge chocolate, chocolate and pistachio, I would choose chocolate since it balances pleasure and pain. Three-choice preferences are not determined simply from two-choice preferences—context matters.

Intransitivity arises in these cases because my preferences are based on two competing attributes, which form a partial rather than a linear order. A linear order exhibits transitivity by definition. Several authors have studied various partial orders in order to study the concept of intransitivity. Some authors consider weak orders [53], which include an equivalence relation known as incomparability or indifference. Luce considered semiorders [53] in which indifference was allowed to be intransitive and where the ordering is induced by a utility function. Fishburn formally studied intransitivity [54] in a wide variety of settings and suggested that transitivity is not essential to ensure the existence of maximally preferred alternatives in a number of situations. Butler and Progrebna [44] studied the Steinhaus and Trybula paradox, in which the probabilities of the choices all exceed 50% and yet weak stochastic transitivity (WST) is still violated; based on a number of experiments, they concluded that “Results support our conjectures that the cycles reflect latent intransitive preference rather than noisy implementation of transitive preferences.” These experiments demonstrated that people use different strategies in different contexts; a strategy which appears rational in one context may not be so in a different context, thus making human decisions adaptable to different circumstances. Makowski et al. [55] presented a simple two-player choice game and showed that the optimal strategy of one player can only be intransitive while that of the second player may be transitive or intransitive. In a quantum version of the game, it turns out that that there is a certain course of the game where only intransitive strategies are optimal for both players. Klimenko [56] examined intransitivity across a wide range of settings and concluded that intransitivity should appear under any of the following conditions: relative comparison criteria, multiple incommensurable comparison criteria, multiple comparison criteria that are known approximately or comparisons of groups of comparable elements.

### 1.2. Intransitivity in Collective Intelligence Systems

Collective intelligence refers to adaptive, intelligent behavior [57] which arises in the absence of any central authority, control or planning. A collective intelligence system is able to make ecologically salient choices in response to changing environmental conditions or contexts through the collective action of a large numbers of agents, none of which hold any authority or complete knowledge of the situation. In nature, the prototypical example of collective intelligence is a social insect colony such as that of ants, social bees and wasps [58,59,60].

Individual workers of a colony often exhibit nonrational decision making such as intransitive preferences. For example, workers of the wasp species *Vespula germanica* were observed individually searching for food [61]. The wasps appeared to utilize two attributes to determine their search preferences: the quantity of the food and the location of the food. Prior experience influenced which attribute was dominant, with the quantity of the food dominating initially but location dominating subsequently. Thus, in some situations, irrelevant contextual information (the location) resulted in intransitive behavior [61]. Honeybee workers also exhibit intransitivity in preferences [62]. Presented with a set of artificial flowers which varied in height and sucrose concentration, from A (short and weak) to D (long and high), the workers exhibited an intransitive preference in the order of A > B > C > D < A. In fact, they found workers which violated weak stochastic transitivity and others that violated strong stochastic transitivity. Attempts to teach workers of *Apis mellifera* a transitive preference hierarchy failed, seemingly due to memory constraints rather than contextual effects [63].

The decoy effect provides another example of nonrational preferences. The decoy effect occurs in a setting in which preferences are based on two attributes. A subject is presented with two alternatives, neither of which is clearly superior, and an asymmetrically dominated decoy option (meaning that it is inferior to (dominated by) one option but not by the other). In the absence of the decoy, neither alternative should be preferred. In the presence of the decoy, the subject will show a preference for the dominating option. This violates the principle of regularity, which asserts that a preference should not change simply due to the presence of additional, nonpreferred options. Workers of *Apis cerana* (Asian hive bees) [64] and *Apis mellifera* (honeybees) [65] have been observed to exhibit the decoy effect. Latty and Trueblood [66] argue that flower choice is a complex process involving multiple considerations: economics, constancy, choice-set size, innate preferences and composition, and these in turn are influenced by attributes such as sex, age, nutritional state, satiation and experience. The belief that preferences should form a linear transitive order seems unduly simplistic in the face of such complexity.

Workers of the ant species *Temnothorax albipennis* also make preference decisions concerning nest sites based on multiple attributes such as the level of lighting, entrance size and height. They too exhibit the decoy effect [67,68].

Workers of the ant species *Lasius niger* are influenced by additional contextual effects. For example, they may judge food quality relative to some reference point rather than based upon some absolute value [69]. They may be influenced by the presence of labels such as odor [70,71]. Workers of the ant species *Atta cephalotes* appear to modify their food preferences in response to the variable abundance of potential food, favoring whichever leaf is less abundant [72]. Workers of the ant species *Atta insularis*, which are indifferent to two different exits, may break this symmetry in the presence of alarm pheromones and preferentially seek just one exit (though the choice appears random) [73].

More interesting is the behavior of the social insect colony as a whole; the collective intelligence of the colony as opposed to the intelligence of individual workers. Individual workers are capable of complex decision making, taking account of a potentially large number of factors and integrating those assessments into a single choice, which may not conform to the restrictions of rationality but may nevertheless provide the resilience, adaptability and robustness necessary for survival. The concept of a naturally occurring computational system (NOCS) [74,75] makes explicit the distinction between decision making carried out by a living agent in a complex environment with imperfect knowledge and on the fly and the idealized agents presented in theory, which exist in simplistic, unchanging environments and have perfect information and infinite time to search out optimal choices relative to some arbitrary (often conjectured) criterion. Resilience, robustness and adaptability are often much more important than optimality [74,75]. The dynamics of collective intelligence systems (and NOCS generally) is characterized by generativity, transience, mass action, interaction, emergence, contextuality, openness to the environment, stigmergy, creativity, symmetry breaking and many other properties [2,76,77].

Franks and colleagues carried out detailed studies of nest emigration by colonies of *Temnothorax albipennis* [78]. They carried out detailed observations of nest emigration including tracking the movements of each worker individually. When emigration is forced, for example by destroying the nest, workers stream out and explore the surrounding environment. Each worker examines a potential site, and if an individual threshold is exceeded, they return to the original nest site to recruit fellow nestmates, eliciting tandem running or carrying. O’Shea-Wheller et al. [79] observed that individual workers appear to manifest a heterogeneous range of decision thresholds, which manifest in the duration that they spend in a potential nest site. Using a computer simulation, they showed that the presence of heterogeneous thresholds allowed the colony to effect optimal, self-organized emigration decisions without the need for direct comparisons by individual workers. Franks showed that if a quorum threshold is exceeded by returning workers (a form of plebiscite), then the majority of nestmates will begin emigration to the site chosen by the majority [78].

Franks and colleagues [78] found that colonies form preference hierarchies based on multiple attributes and, under some circumstances, are able to utilize a weighted additive decision strategy, which is difficult even for humans. Decisions at the colony level may sometimes be rational when those of the individual workers are nonrational. Edwards and Pratt [67] showed that colonies of *Temnothorax* avoided falling prey to the decoy effect even though their workers individually did not. Contextual factors play a role here. A study of foraging by *Myrmica rubra* [80] found that modifying the available choice set by increasing the number of nest entrances from one to two resulted in worse foraging outcomes. In another study [81], prior exposure to an alternative nest influenced the choice of nest site on subsequent emigration. Using formal modeling, the authors showed that workers need not utilize comparative strategies to effect decisions, as is often assumed to be the cause of nonrational decisions. While comparative strategies might manifest at the colony level, individual workers could use absolute strategies combined with threshold-based decision rules, demonstrating how an experience-dependent, flexible strategy can emerge at the global level from a fixed-threshold strategy at the local level. Doran et al. [82] found that the tendency of a colony to move was not based on the value of alternate sites in some abstract sense but instead based on the potential fitness benefit of moving. In an already good nest site, no migration would convey significant fitness benefits, but for a colony recently made homeless, any nest would do. Context dependency suggests that two nest sites assessed under different conditions may be evaluated by using different preference hierarchies.

The speed of search may depend upon prior living conditions, being faster if forced from a good nest than from a poor nest, suggesting the presence of an urgency hypothesis [83]. If workers are exposed to an alternative nest site of lower quality than their own and then forced to emigrate facing the familiar alternative nest and a novel alternative of similar quality, they tend to avoid the familiar site and opt for the novel, breaking preference symmetry. However, if presented with familiar and novel high-quality sites, they maintain symmetry.

The study of decision making among social insects has proven to be a fruitful subject matter for the application of sophisticated mathematical and computational models [76,77], particularly in the past 20 years. Houston [84] showed formally that the fitness value of any food item was contextual rather than absolute, dependent on its alternatives and its probability of being foraged. Nicolis et al. [85] pointed out that collective intelligence systems often rely upon some form of positive feedback in order to effect their decision making. They showed that, generically, the probability of choosing the best out of a choice of n options depended crucially upon the strength of the feedback. There is an optimal level of feedback which maximizes this probability, and this optimal value of feedback depends upon the number of options. Sasaki et al. [86] modeled *Temnothorax rugatulus* and showed that colonies outperform individuals when the degree of difference between the options is small, so discrimination is difficult. When the degree of difference is large and discrimination is easy, individuals outperform colonies.

### 1.3. Obstacles to Forming Cyclic Systems from Available Data

The literature on decision making by social insect colonies was examined to see if there were any results that could be examined for the presence of true or Type II contextuality. There is certainly abundant evidence in support of Type I contextuality and intransitivity. Some studies [73] demonstrated intransitive preferences but examined only one context and thus are noncontextual. Some studies examined many contexts but only measured one object (for example, [69]), so any cyclic system could have many bunches, but each bunch would be composed of one object and so would not in fact be a cyclic system. Again, this cannot be used to test for contextuality. Many studies with multiple objects and contexts cannot form cyclic systems. For example, Oberhauser [70] studied two objects (the probability of motion towards a sucrose solution) and in two contexts, one where the ants had conflicting information about the location of the 1.5 M sucrose solution and one where they did not (context 2). However, Oberhauser et al.’s study did not form any cycles.

Unfortunately, the problem of mutually exclusive outcomes preventing the formation of cyclic systems is the rule, not the exception, in studies of intransitive behavior by ants (for example, [67,79,81,87,88]).

### 1.4. Previous Work: A Cyclic System of Rank 3

In a previous study [37], we examined data from a study previously published by Doran et al. [82]. It was not possible to form a 4-cyclic system, but it was possible to form a 3-cyclic system with these three objects:q1= marginal probability of emigration from a low-quality nest within 6 h.q2= marginal probability of emigration from a mediocre nest within 6 h.q3= marginal probability of emigration from a good nest within 6 h.and three contexts:c2= presence of a mediocre alternative nest.c3= presence of a good alternative nest.c4= presence of an excellent alternative nest.

Unfortunately, in this case, ΔC=−1.99, showing no sign of true contextuality or Type II contextuality, although it does exhibit Type I contextuality.

## 2. Materials and Methods

Most of the experiments in the literature were conducted without any concerns about their structure as cyclic systems. Indeed, the issue of true or Type II contextuality is not mentioned in these experiments. Thus, we decided to create a simulation-based study by using one of the excellent simulations developed by Robinson (with their kind consent) and reported in the literature [81]. We created a rank-four cyclic system, which consists of four objects and four contexts, giving rise to eight random variables as depicted in the content–context matrix (set *n* = 4):
$q_{1}$$q_{2}$$q_{3}$⋯$q_{n-1}$$q_{n}$
$R_{q_{1}}^{a_{1}}$



$R_{q_{n}}^{a_{1}}$$a_{1}$$R_{q_{1}}^{a_{2}}$$R_{q_{2}}^{a_{2}}$



$a_{2}$
$R_{q_{2}}^{a_{3}}$$R_{q_{3}}^{a_{3}}$


$a_{3}$



⋱
⋮



$R_{q_{n-1}}^{a_{n}}$$R_{q_{n}}^{a_{n}}$$a_{n}$


Robinson’s model examines the behavior of ants scouting for new nest sites. This model involves Monte Carlo simulation methods and treats the ants as separate realizations of a Markov process. It assumes that ants evaluate sites based on a simple threshold rule, where each ant has a certain minimum quality nest site it will recruit to. Furthermore, the model asserts that these acceptance thresholds are normally distributed across the population of ants. Consequently, the ability of ants to choose the correct nest at the colony level arises not from the recruitment latency but from the fact that the proportion of ants willing to recruit to a good nest will always be higher than the proportion of ants willing to recruit to a poor nest [81].

The Markov process identifies five different states: ‘evaluating home site,’ ‘evaluating inferior site,’ ‘evaluating superior site,’ ‘committed to inferior site,’ and ‘committed to superior site.’ Ants, once committed to a site, cannot switch to a better alternative. However, ants can switch between evaluation states. All ants start in the ‘evaluating home site’ state. However, the home site (with a value of −1000) is uninhabitable. Therefore, the ants will never commit to it [81]. After leaving the home nest, the ants will arrive at alternative nest sites whose locations and environmental features are captured in terms of discovery probabilities and required travel time. These attributes allow the simulation to accurately assess the effect of the distance from the home nest on the probability of recruiting a nest. When a site is discovered, the ant evaluates it with some error, intending to replicate the noise present in the decision making of real ants. If the evaluated quality surpasses the ant’s acceptance threshold, it commits to the site. Otherwise, it continues to search for a better site [81]. The model was tailored to replicate a specific experimental setup and showed compatibility with the observed data. The switching behavior demonstrated in the simulations mirrored the empirical data from Robinson et al.’s 2009 experiment [89], with the majority of ants that found the superior site first remaining there, while those discovering the inferior site differed depending on their individual acceptance thresholds [81]. The model also successfully replicated the empirical finding that the recruitment latency does not significantly impact the results. Despite the quantitative differences (which appeared because the simulated ants could not make tandem runs), the general patterns were consistent between the simulated and empirical data. Furthermore, the model exhibited predictive validity by accurately forecasting the outcome in a novel scenario involving four potential nest sites [81]. This shows that the model not only fits well with historical data but is also capable of making accurate predictions about novel scenarios, reflecting its robustness and utility in understanding ant behavior.

Considering the capacity of Robinson et al.’s simulation to mimic real-world results, we departed from the parameters of their initial experiment as little as possible. The specific setup in question involved simulating a forced emigration from a destroyed nest where the ants were forced to choose between two possible nest sites, one that was nearby (but poor quality) and one which was further away (and high quality) [81]. We differentiated our eight experiments from each other by modifying only one model parameter, the site quality. In Robinson’s model, the site quality is assigned a somewhat arbitrary numerical value. Ants in turn are assigned a threshold value that determines what minimum site quality they will recruit to. Therefore, sites of differing quality can be distinguished in terms of their distance from the mean threshold value for the whole population of ants. When that mean threshold value is set to 5 (as it was in our experiment), a good nest with 1.6 mm walls, a 2 mm wide entrance and a dark interior would be assigned the value 6. A somewhat worse nest, with 0.8 mm walls, 4 mm wide entrances and a dark interior would be assigned the value 4 [81]. We departed only slightly from these values, with the worst nest in our setup being assigned the value 3.1 and the best nest being assigned the value 6.5. Other nests had values between these lower and upper bounds. As a result, all the nests we used had reasonable values and could plausibly describe real nests. The parameter values choseon in our model are given in Table 1, Table 2 and Table 3.

In our setup, the four objects of measurement were the probabilities associated with migrating to each of the four nest sites simulated as being far away from the ant colony. These nest sites were assigned the following values within the simulation:$q_{1}:5$.$q_{2}:5.5$.$q_{3}:6$.$q_{4}:6.5$.

Furthermore, the contexts in our setup were merely the alternative nearby nest sites presented. These nest sites had the following values:$c_{1}:3.1$.$c_{2}:3.6$.$c_{3}:4.1$.$c_{4}:4.6$.

Before running the simulation by using Robinson et al.’s computational model, we tested whether such an experimental setup could conceivably give rise to contextuality by using Masuda’s differential equation-based model of ant-nest-site selection [90] (with their gracious permission). Masuda’s differential equation model examines nest selection at the colony level where ants choose between two potential sites, one good and the other mediocre. The simulation begins with the ants in their current nest. Some fraction of the ant population (determined by the parameter z) will begin scouting for a new site. However, as the model simulates an infinite population, the z parameter affects the duration of but not the ultimate outcome of the simulation. Like Robinson’s model, Masuda’s model assumes that ants have fixed preferences and will recruit only to nests that are above some minimum threshold of acceptability. Consequently, the model bridges the gap between the probability that some individual ant moves to a nest site (the parameter H) and the probability that an idealized ant colony with an infinite population will select a given nest site [90].

The speed at which a colony selects the superior nest site is affected by two additional parameters: $\alpha_{s}$ (the rate at which an ant moves to the superior nest site) and α (the leakage rate: the probability of the ants returning to the home nest) [90]. Neither of these two parameters affect the choice of the nest. As a result, the leakage rate was left unchanged so as to avoid producing less realistic values for the duration. The parameter values chosen for this model are given in Table 4.

Regarding the goodness of fit, the model is highly idealized and operates under the assumption of an infinite population of ants. Consequently, it produces binary values as an infinitely large colony will select a site 100% of the time even if only 51% (or 50.00…1%) of ants prefer it to the alternative [90]. This creates a zero-noise condition, which provides a good illustration of whether contextuality could be violated in theory but should not be mistaken for a realistic result. Therefore, the model provides insight into what might occur in a theoretical, noise-free scenario.

On account of producing binary outcomes, calculating joint probabilities is trivial as the joint probability of any value and 0 is 0. Similarly, if a simulated set of ants selects a particular nest under one condition with a probability of 1 and selects that same nest under another condition with a probability of 1, the resultant joint probability is also equal to 1. Predictably, it is also not meaningful to conduct error calculations when dealing with deterministic simulations.

Calculating the joint probability of an ant accepting the better nest site in both contexts where it was presented was slightly more complicated when dealing with Robinson’s model. Thanks to their fixed thresholds, a subset of ants exists that invariably rejects the inferior nest site in every context. These ants, with high thresholds, will leave poorer quality nests irrespective of when they encounter them. Therefore, the minimum value of the joint distribution can be calculated by determining the minimum number of ants who rejected the poor-quality nest in both trials. This figure represents the population of ants with such high-quality thresholds that they have no alternative to choosing the better site. For the remaining ants, the joint distribution can be computed in a straightforward manner like any independent probability. The ants with lower thresholds will settle in the first nest they come across. Therefore, to calculate the remainder of the joint distribution, we subtract the number of high-threshold ants from the total number of ants that accepted the good site in both experiments and multiply the results of the two experiments: (accept good in experiment 1 − high threshold) * (accept good in experiment 2 − high threshold). Adding these two values together provides a joint distribution. The error for the probability of accepting the good site is simply measured in terms of the standard error of a proportion. The joint distribution involved dependent probabilities, and consequently the error was calculated by using bootstrapping with 100,000 resamples for each of the four joint distributions. The calculations of contextuality introduce no new errors, so all the errors presented in those calculations were downstream of the errors in the probabilities and joint distributions of the simulations themselves.

We analyzed the data resulting from the two simulations by using Dzhafarov’s modified version of the CHSH inequality as described above [20].

## 3. Results

Masuda’s differential equation model revealed that our experimental setup could come close to demonstrating contextuality under conditions with no noise. Our simulation almost violated the CHSH inequality with ΔC=2 and almost violated Dzhafarov’s version of the inequality with ΔC=0. The results from Robinson et al.’s model proved less promising. The ants consistently demonstrated preference reversal. When presented with a choice between a good but distant nest and a nearby one, they consistently choose the better of the two options. However, when presented with the nest $q_{3}$ in context 4, they departed from their normal behavior and chose the worse of the two nests. Nonetheless, the results violated neither the CHSH inequality (ΔC=−1.68±0.0482) nor Dzhafarov’s version of the inequality (ΔC=−2.87±0.0538).

## 4. Discussion

Our results demonstrate that in an idealized noise-free system, the behavior of ants, in particular their susceptibility to preference reversal, could conceivably violate even the inequality proposed by Dzhafarov. These results did not appear in the more realistic simulation for a number of reasons. To begin, neither the ants’ initial preferences nor their eventual preference reversal were particularly strong. For example, in all experiments, excepting the one involving preference reversal, the ants only accepted the good nest in between 50% and 80% of the trials. The eventual preference reversal was comparably small, with ants still accepting the better of the two available nests in 49% of the trials. The consequence of this is that the negative term in the CHSH inequality was not sufficiently negative enough to generate contextuality. This issue could conceivably be avoided if multiple drivers of intransitive behavior were combined in a single experiment, which could produce a starker preference reversal. However, the weakness of the preference reversal was not the chief obstacle to producing contextuality. More troublingly, the CHSH inequality depends in part on the joint probability of the ants accepting the good nest in both contexts where it is present. However, although the ants in the simulation have fixed preferences and are thus more likely than chance to choose the same nest in successive contexts, their nest selection behavior also depends on the probability of discovering the same nest in both experiments. Consequently, the joint probabilities ranged from 0.31 to 0.51. Due to these lower-than-ideal joint probabilities, the positive terms in the CHSH inequality were not sufficiently positive to generate contextuality. This problem of joint probabilities is likely an intrinsic property of ant behavior. While natural systems may be too noisy to exhibit true contextuality, it is possible that artificial collective intelligence systems might exhibit true contextuality, and its study may provide some illumination of the conditions under which true contextuality could occur in neural systems.

## 5. Conclusions

Type II contextuality has been claimed as an exclusive characteristic of quantum systems. A number of studies have demonstrated the presence of Type II contextuality in psychological experiments with human subjects. It is argued that intransitivity in decision making may play a role in the appearance of Type II contextuality. Thus, Type II contextuality was sought in the decision-making behavior of social insect colonies, where intransitivity often appears. An example of collective intelligence was presented, in which cooperation and the intransitivity of preferences was present, and through computer simulations, a search was made for evidence of Type II contextuality. Unfortunately, this was not found, but only a limited sample of situations could be explored. Further research is indicated, and it would be ideal if such behavior could be observed in living collective intelligence systems.

## Figures and Tables

**Table 1 entropy-25-01193-t001:** Parametrization used in Robinson et al. [89].

Parameter	Value	Derivation
Number of nests	3	Robinson et al. [89]
Position of nests	Good nest further than poor nest	Robinson et al. [89]
Mean travel time between nests	$\begin{array}{cccc} & Old & A & B\\ Old & 1 & 36 & 143\\ A & 36 & 1 & 116\\ B & 143 & 116 & 1 \end{array}$	From walking speed of 8.4 mm/sec data
Probabilities of finding nests	$\begin{array}{cccc} & Old & A & B\\ Old & 0.91 & 0.15 & 0.03\\ A & 0.06 & 0.80 & 0.06\\ B & 0.03 & 0.05 & 0.91 \end{array}$	Robinson et al. [89]
Number of ants	10,000	Arbitrary
Acceptance threshold Distribution	Normal, mean = 0, SD = 1	Arbitrary
Nest qualities	See Table 2	Arbitrary
Assessment error	Normal, mean = 0, SD = 1	Arbitrary

**Table 2 entropy-25-01193-t002:** Parametrization used in Robinson et al.’s [89] simulation (cont’d).

Experiment Number	1	2	3	4	5	6	7	8
Value of old nest	−1000	−1000	−1000	−1000	−1000	−1000	−1000	−1000
Value of poor nest	4.6	3.1	3.1	3.6	3.6	4.1	4.1	4.6
Value of good nest	6.5	6.5	5	5	5.5	5.5	6	6

**Table 3 entropy-25-01193-t003:** Parametrization used in Robinson et al.’s [89] simulation (cont’d).

Experiment Number	1	2	3	4	5	6	7	8
Probability of accepting good site	0.50±0.00844	0.79±0.00976	0.78±0.00973	0.68±0.00940	0.69±0.00943	0.58±0.00943	0.59±0.009	0.49±0.00836
Rate of switch to good nest	0.18	0.46	0.50	0.40	0.38	0.28	0.27	0.18
Duration (min.)	131	206	497	526	247	231	168	163
Joint probability of accepting good nest in both contexts	0.38±0.00345	0.38±0.00345	0.51±0.00327	0.51±0.00327	0.40±0.00322	0.40±0.00322	0.31±0.00326	0.31±0.00326

**Table 4 entropy-25-01193-t004:** Parametrization used in Masuda et al.’s [90] differential equation simulation.

Parameter	Value	Derivation
Rate of switch to good nest $\alpha_{s}$	See Table 3	From Robinson et al.’s simulation [89]
Leak rate α	0.1	Arbitrary
Portion of scouts *z*	0.2	Arbitrary
Probability of acceptance of good nest *H*	See Table 3	From Robinson et al.’s simulation [89]

## Data Availability

Data are available from the authors (W.S. and A.K.).

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
