# Peer review of "Contextuality in Collective Intelligence: Not There Yet"

_entropy, 2023, doi:10.3390/e25081193_

Round 1

Reviewer 1 Report

This is the novel is stimulating field of research and it is not surprising that the authors have not yet completely achieved the aim. One of casualties is that it seems that they do not know about the paper  

Journal of Experimental Psychology: General 148, 1925-1937,

which closely related to the paper under review. I strongly recommend the authors to read this paper and may be this will help to improve the result. I also suggest to mention the early papers on Bell inequality for cognitive systems, as  

Conte, E., Khrennikov, A., Todarello, O., Federici. A., Mendolicchio, L.,  and  Zbilut, J. P.:  

A preliminary experimental verification on the possibility of Bell inequality violation in mental states. 

NeuroQuantology  \textbf{6}, 214-221 (2008).

 Asano, M.,  Khrennikov, A., Ohya, M., Tanaka, Y., and   Yamato, I.: Violation of contextual generalization of the Leggett-Garg inequality for recognition of ambiguous figures. Phys. Scr.    \textbf{T163}, 014006 (2014).

Author Response

Thank you very much for your kind review. I was aware of one of the papers mentioned - that regarding True Contextuality, and omitting it was an oversight. The other two papers were unfamiliar to me - I have included them in the references as well. Thank you for helping my paper to be more inclusive.

Reviewer 2 Report

This paper deserves the highest ratings due both to its specific results and their implications for the form of knowledge in general about the relation between quantum and non-quantum systems. The authors - and this is not as common as it should be - make sure to give "equal time" to the latter.

The next step that I hope the authors would take is to look at non-collective non-quantum instances of intransitivity for example in "simple" human relations. The "Logic in Reality" proposed by Brenner provides an explanatory concept that insures that the discussion remains within the bounds of natural science. As Brenner states, this logic is clearly non-Kolmogorovian.

The Dzhaforov inequality might not apply to such systems, but that would be a limitation of mathematics not of the openness of living systems. I also recommend here the recent systems work of Gianfranco Minati in Milan. 

The paper should be published without question.

Author Response

Thank you very much for your kind review. I very much appreciate the reference to Brenner, whose work I was not familiar with. IT does indeed look very relevant to the work that I am doing and I look forward to reading it. I added a comment in this regard to the acknowledgments.

Round 2

Reviewer 1 Report

accept as it is